# Why women choose to give birth at home: a situational analysis from urban slums of Delhi

Niveditha Devasenapathy,[1] Mathew Sunil George,[1] Suparna Ghosh Jerath,[1] Archna Singh,[2] Himanshu Negandhi,[1] Gursimran Alagh,[1] Anuraj H Shankar,[3] Sanjay Zodpey[1]

For numbered affiliations see end of article.

**Correspondence to**
Dr Niveditha Devasenapathy;
niveditha@iiphd.org

## ABSTRACT

**Objectives:** Increasing institutional births is an important strategy for attaining Millennium Development Goal -5. However, rapid growth of low income and migrant populations in urban settings in low-income and middle-income countries, including India, presents unique challenges for programmes to improve utilisation of institutional care. Better understanding of the factors influencing home or institutional birth among the urban poor is urgently needed to enhance programme impact. To measure the prevalence of home and institutional births in an urban slum population and identify factors influencing these events.

**Design:** Cross-sectional survey using quantitative and qualitative methods.

**Setting:** Urban poor settlements in Delhi, India.

**Participants:** A house-to-house survey was conducted of all households in three slum clusters in north-east Delhi (n=32 034 individuals). Data on birthing place and sociodemographic characteristics were collected using structured questionnaires (n=6092 households). Detailed information on pregnancy and postnatal care was obtained from women who gave birth in the past 3 months (n=160). Focus group discussions and in-depth interviews were conducted with stakeholders from the community and healthcare facilities.

**Results:** Of the 824 women who gave birth in the previous year, 53% (95% CI 49.7 to 56.6) had given birth at home. In adjusted analyses, multiparity, low literacy and migrant status were independently predictive of home births. Fear of hospitals (36%), comfort of home (20.7%) and lack of social support for child care (12.2%) emerged as the primary reasons for home births.

**Conclusions:** Home births are frequent among the urban poor. This study highlights the urgent need for improvements in the quality and hospitality of client services and need for family support as the key modifiable factors affecting over two-thirds of this population. These findings should inform the design of strategies to promote institutional births.

### Strengths and limitations of this study

- This survey covered a large number of households (n=6092) living in three urban poor settlements of Delhi.
- Qualitative and quantitative methods were used to capture reasons for home births.
- Although the slum cluster was not a random sample from all the slum clusters of Delhi, it was representative of the urban poor settlements.
- Concurrent health facility assessment was not performed which would have helped to understand additional supply-side factors.

## INTRODUCTION

Increasing institutional births is a key global strategy to reduce maternal and new-born mortality. Many countries, including India, have established incentive programmes and policies to enhance institutional births. However, the rapid growth of low-income urban population presents unique challenges to these programmes, such as lesser knowledge of local services and registration processes, lack of support from an extended family and transient residence. With around 40% of urban population in low-income and middle-income countries residing in low-income urban settlements, more focused efforts are required to improve institutional birth rates in these settings.

Most maternal deaths are centred around the intrapartum and immediate postpartum period[1 2] and for countries with a high burden of maternal mortality and morbidity, facility-based birthing is found to be efficient and sustainable compared with scaling up community-based safe birthing programmes.[3 4] An evaluation study of the safe motherhood programme in Indonesia showed that irrespective of level of socioeconomic status and place of

residence (urban vs rural), increasing the number of deliveries by skilled birth attendants (SBAs) did not reduce maternal mortality if most births took place at home.[5] In contrast, programmes that enhanced facility-based births in Malaysia and Sri Lanka resulted in marked reduction in maternal and neonatal deaths.[6] With overwhelming evidence in favour of scaling up quality health centre-based intrapartum care to improve maternal and neonatal survival, it is important for each country and local administration to understand the barriers at community and facility levels that affect access to facilities and provision of quality services.

India currently accounts for about a fifth of all maternal and new-born deaths worldwide,[7] and approximately one-third of the population currently lives in urban areas, which is going to be nearly one-half by 2030. The MDG report has flagged the slow progress of India in reducing child mortality and improving maternal health,[8] with the latest WHO statistics showing that only half of expectant mothers in India complete four antenatal care (ANC) visits and give birth in the presence of a SBA.[9] Overcoming barriers to institutional births among the urban poor is, therefore, crucial.

The National Rural Health Mission (NRHM) of India launched the Janani Suraksha Yojana (JSY) programme in 2005 with the goal of reducing maternal and neonatal mortality by promoting institutional births among poor pregnant women.[10] An evaluation of this conditional cash transfer scheme in 2007–2008 showed an increase in ANC visits and institutional births.[11] However, this has not translated into reduction in MMR possibly due to unaddressed issues of non-financial access barriers and suboptimal ANC, delivery and postnatal care.[12] In addition, unique issues faced by the urban poor were not specifically addressed by the JSY programme.

As aforementioned, currently 30% of the Indian population is living in cities. Delhi is one of the most densely populated cities in the world, and attracts nearly 500 000 migrants every year with most settling in urban poor habitations.

According to the National Family Health Survey 3 (NFHS 3) conducted in 2005–2006, only 44% of births were institutional among the urban poor of Delhi as compared with the urban average of 67.5%.[13] The District Level Household and Facility Survey (2007–2008) showed that overall 71% of pregnant women had at least three ANC visits. While 68% of births were institutional in the city as a whole, only 38% institutional deliveries were reported in slum areas.[14] A governmental initiative aimed at correcting this inequity is the National Urban Health Mission (NUHM) which makes primary healthcare services available to the urban poor.[15 16] The success of this mission will depend on identifying and targeting interventions directed towards the most vulnerable. One of the aims of this study is to determine the prevalence of home and institutional births among women living in urban poor settlements in Delhi and identifying reasons influencing their choice.

## METHODS

This study is part of the formative phase and situational analysis for the ANCHUL (Ante Natal and Child Healthcare in Urban Slums, CTRI/2011/091/000095) trial, an implementation research project aimed to develop, implement and evaluate the impact of an intervention package delivered through an urban community healthcare worker (UCHW). ANCHUL aims to increase access to healthcare facilities for birthing and improve maternal, neonatal, child healthcare (MNCH) practices in urban slums of Delhi. This study aimed to conduct an in-depth situational analysis on the utilisation and quality of MNCH care using quantitative and qualitative methods. The information obtained will guide the development of the community-based intervention package to be delivered by the UCHW as part of the ANCHUL trial.

### Study setting

Of the 16.7 million people living in Delhi, 52% reside in poor habitations.[17] The north-east district of Delhi contributes 11%, that is, about one-fifth, to this population with 44 registered slums.[18] This district has the highest home birth rate.[14] We conducted a rapid survey in 17 slum clusters to obtain information on the number of households, water supply, sanitation, presence of schools, healthcare facility and distance from nearest government hospital. The clusters were then stratified into two categories of vulnerability based on the above characteristics. We then randomly selected three vulnerable slum clusters namely Buland Masjid, CPJ and Chanderpuri (CP). These slums had tarred roads and had access to maternal child healthcare dispensaries within a distance of 5 km. The study protocol was approved by the Health Ministry Screening Committee of the Government of India, institutional ethics committees of the Public Health Foundation of India, All India Institute of Medical Sciences, WHO Geneva and Harvard School of Public Health.

### Data collection
#### Quantitative survey

After lane mapping the clusters, all households were included in the survey. We identified pregnant women (in their second and third trimesters), recently delivered women (RDW, ie, those who had delivered in the past 3 months) and households with under-5 children. The purpose of the survey was explained to a household member above 18 years of age and all questionnaires were administered after obtaining informed consent by trained field interviewers. All survey tools were in local language and were piloted and modified for content and clarity. Information on family details, sociodemographic status, place of childbirth (in women who had given birth in the past 1 year) and information on any maternal and child deaths within households in the past 1 year was obtained using paper forms. All refusals and non-responses were documented. We revisited the households of 160 RDW and collected detailed

information about ANC, delivery, immediate postnatal period, new born care practices and diet of the mother. Data were checked for completion before entering into a structured database management system (Microsoft Access 2010) with inbuilt range and internal consistency checks. Information from RDW was edited and validated by double data entry.

### Qualitative data

The categories of respondents in table 1 were identified as relevant for data collection in this study. Households were informed that focus group discussions (FGDs) would be held in the community and a general invitation was given. Permissions were sought from local community and religious leaders. Local public and private healthcare providers were approached and permissions were sought for in-depth interviews (IDIs). The FGD and IDI guides were piloted to refine the topic guides to enable them to generate data that were relevant to the study objectives. The main topics that were explored in the FGDs and interviews are shown in table 1. The venue for data collection was agreed on based on the respondents' convenience. One interviewer facilitated the discussions while a second took notes. Based on responses from the community, healthcare facilities and traditional birth attendants who served the locality were identified and approached. Written informed consent was obtained from all participants before the FGDs and IDIs, which were digitally voice recorded.

### Sample size justification

For estimating the number of households to be interviewed, institutional birth was considered the key outcome variable. Assuming prevalence of institutional births as 33% in urban slums of Delhi,[16] we need to interview 780 women who gave birth in the past year to obtain prevalence estimates with 10% relative precision. Assuming a crude birth rate of 25/1000 (national average is 21/1000), a population of 30 000 was to be covered to identify at least 750 childbirths in the past year.

### Data analysis

#### Quantitative data

Data were analysed using Stata V.11 (Stata Corporation, College Station, Texas, USA). Descriptive statistics were used to provide cluster, household and individual level profiles of the study population. Household survey data were analysed accounting for clustering at the slum level to control for intercluster and intracluster variance. We used principal component analysis to compute household socioeconomic scales (SES). Dwelling characteristics, household income and household assets were included in this composite scale.[19] We used multivariate random effects logistic regression to estimate the association of demographic variables with home births. Crude and adjusted ORs were calculated with 95% CIs. For data from RDW, Pearson $\chi^2$ was used for categorical variables and student t tests for comparison of continuous variables.

#### Qualitative data

Verbatim transcripts were prepared in a standardised format that included basic demographic information of the participants and the interviewer's own observations within 1 week of conduct of IDI/FGD. Transcripts were uploaded to a software Atlas ti V.6.1 (Scientific Software

| Table 1 Themes covered for qualitative data | | |
|---|---|---|
| **Category of participants** | **Method of data collection** | **Key themes covered** |
| *Community*<br>▸ Pregnant women (n=5)<br>▸ Recently delivered women (n=6)<br>▸ Mother of under 5 children (n=6)<br>▸ Mothers-in-law (n=5)<br>▸ Husbands (n=4) | Focus group discussions (FGDs)<br>Venue: schools, NGO, madrassa (religious place) and anganwadi centres* | ▸ Health and nutritional status<br>▸ Cultural practices for nutrition during pregnancy<br>▸ Care-seeking behaviour during pregnancy<br>▸ Barriers to accessing care during pregnancy<br>▸ Quality of care experienced in various healthcare settings (public and private) |
| *Healthcare providers*<br><br>▸ Public health system (n=6)<br>▸ Private (n=5)<br>▸ Others (n=4)(AWW, TBAs) | In-depth interviews<br>Venue: clinics of healthcare providers or homes of key informants | ▸ Care-seeking pattern among the community during pregnancy<br>▸ Challenges to improving maternal and child health among the urban poor<br>▸ Feasibility of proposed intervention |
| *ANC clinic attendees* (n=9) | Exit interviews<br>(preANC and postANC check ups)<br>Venue: clinics | ▸ Experience of care during ANC visit<br>▸ Satisfaction levels of the individual about care |

*The word anganwadi means 'courtyard shelter' in Hindi. They were started by the Indian government in 1975 as part of the Integrated Child Development Services programme to combat child hunger and malnutrition.
ANC, antenatal care; AWW, anganwadi workers; TBA, traditional birth attendants *(Dai)*.

Development, City West, Berlin, Germany) and coded line-by-line using detailed themes and subthemes that emerged from the data. After an initial round of coding with a representative sample of transcripts, the list of codes that were generated was reviewed in order to develop a structured code list which was then applied to the remaining transcripts. Illustrative quotations that captured the key issues reported by the participants have been included in the results.

## RESULTS
Of the 6348 households in the three defined clusters, 6092 (96%) were interviewed between December 2011 and March 2012, covering a total population of 32 034. Nine households refused to participate and 247 did not respond (locked houses; figure 1). A total of 25 FGDs and 13 IDIs were conducted in January and February 2012. The number of respondents in each FGD ranged from 7 to 12 members.

### Population and cluster characteristics
The adult male-to-female ratio was 1000 : 825. Fifty-eight per cent of the people were migrants from Uttar Pradesh (73%) and Bihar (16%). Eighty per cent were living in the same locality for >5 years. Of the total population, women of reproductive age (15–49 years) accounted for 25%, and 16.6% were under-5 children. The area was served by 1 referral hospital situated within

a distance of 5 km, 2 outpatient dispensaries, 17 private clinics (registered and unregistered with the Medical Council) and 1 laboratory within the clusters. The areas also have access to two referral hospitals situated at a distance of about 10 km.

### Household characteristics
The median family size was 5 (IQR 4, 7), predominantly living as nuclear families (79.4%) and 63% of houses were self-owned. The head of the household (HOH) was the one considered as the decision maker but was not necessarily the primary wage earner. Fifty-nine per cent of HOHs were illiterate and were unskilled labourers. Ration cards, Below Poverty Line cards (BPL) and Rashtriya Swasthya Bima Yojna (RSBY) cards that are needed for claiming government-run health schemes were possessed by only 50%, 31% and 24% of households, respectively. The majority of households (82%) lived in single-roomed concrete houses with cemented or tiled flooring. Most houses (95%) had access to toilets within the household or community. A detailed sociodemographic profile of the study population is presented in table 2. The household characteristics of the RDW and the overall population in the study area were similar indicating that our subsample households were representative of this area. Fifteen maternal deaths, 21 stillbirths and 41 under-5 child deaths were reported for the previous year. Of the total child deaths, 22 were in the neonatal period.

**Figure 1** Quantitative survey sampling. HH, household.

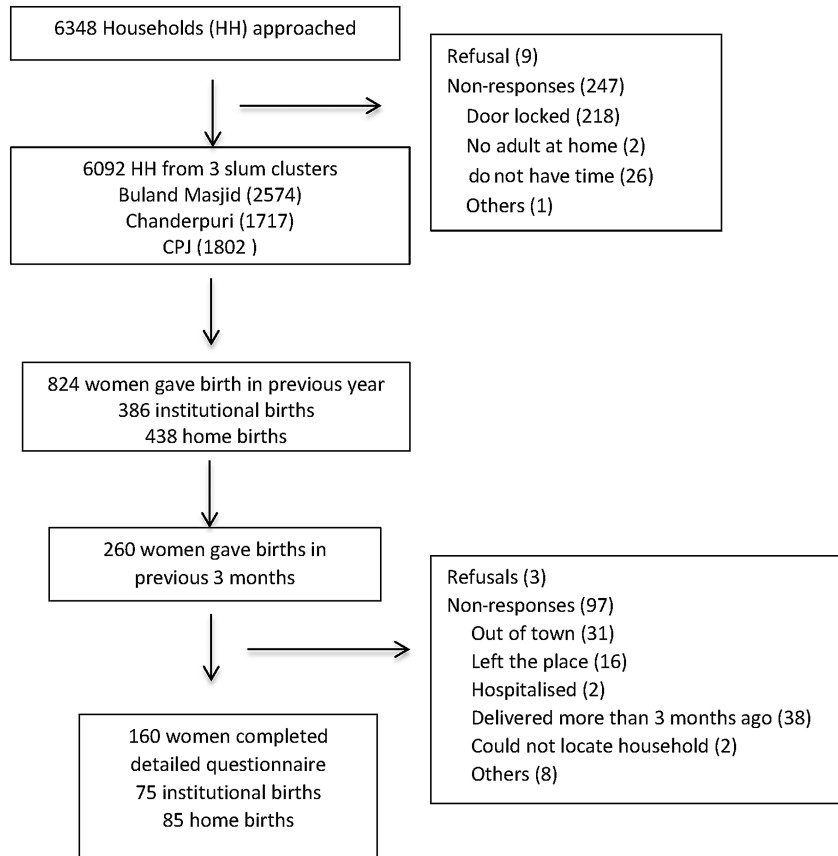

**Table 2** Characteristics of households in the study area and households of women who gave birth in the previous 3 months

| Demographic characteristics | House-to-house survey (n=6092) HH from 3 clusters | Mothers who recently gave birth (n=160) |
|---|---|---|
| Median HH size (IQR) | 5 (4, 7) | 5 (4, 7) |
| Family type (%) | | |
| Nuclear | 4834 (79.4) | 115 (71.9) |
| Joint | 944 (15.5) | 43 (26.9) |
| Extended | 313 (5.1) | 2 (1.3) |
| Spoken language (%) | | |
| Hindi | 5328 (87.5) | 145 (90.6) |
| Urdu | 624 (10.2) | 15 (9.4) |
| Others | 140 (3.2) | – |
| Religion (%) | | |
| Hindu | 1822 (29.9) | 42 (26.3) |
| Muslim | 2475 (69.6) | 118 (73.8) |
| Others | 33 (0.5) | – |
| Caste category (%) | | |
| General | 2546 (41.8) | 77 (48.1) |
| Other backward caste | 2553 (41.9) | 58 (36.3) |
| Scheduled caste/scheduled Tribe | 932 (15.3) | 22 (13.8) |
| Do not want to answer | 4 (0.1) | – |
| Do not know | 57 (0.9) | 3 (1.9) |
| Illiterate women in reproductive age group (%) | | |
| (n=8056) | 4122 (51.2) | 85 (53.1) |
| Literacy level of HOH (%) | | |
| Illiterate | 3561 (58.5) | 93 (58.1) |
| Literate but no formal education | 196 (3.2) | 4 (2.5) |
| Schooling | 2129 (36.4) | 62 (38.8) |
| College | 115 (1.89) | 1 (0.6) |
| Occupation of HOH (%) | | |
| Unskilled | 2805 (46.3) | 77 (48.4) |
| Skilled | 1378 (22.7) | 36 (22.6) |
| Office work | 867 (14.3) | 19 (12) |
| Professional | 55 (0.91) | – |
| Not working | 955 (15.76) | 27 (17) |
| Median HH income in INR (IQR) | 4000 (3000–6500) | 4000 (3000–7000) |
| Median HH income in USD (IQR) | 76.2 (57.1–123.8) | 76.2 (57.1–133.5) |
| Own house (%) | 3829 (62.9) | 101 (63.1) |
| Ration card (%) | | |
| Do not have | 2994 (49.2) | 91 (56.9) |
| White | 1173 (19.3) | 22 (13.8) |
| Yellow | 1196 (19.6) | 28 (17.5) |
| Pink | 686 (11.3) | 18 (11.3) |
| Do not want to answer | 43 (0.7) | 1 (0.6) |
| BPL card (%) | 1903 (31.2) | 47 (29.4) |
| RSBY card (%) | 1461 (24) | 36 (22.5) |
| Per cent of HH who are staying in the current locality in years | | |
| <1 | 660 (10.8) | 13 (8.1) |
| 1–2 | 238 (3.9) | 11 (6.9) |
| 3–5 | 330 (5.4) | 12 (7.5) |
| >5 | 4864 (79.8) | 124 (77.5) |
| Per cent belonging to Delhi | 3572 (57.49) | 93 (58.1) |
| Per cent of HH who migrated but living in Delhi in years | | |
| <1 | 261 (7.5) | 2 (2.2) |
| 1–2 | 123 (3.5) | 3 (3.2) |
| 3–5 | 195 (5.6) | 2 (2.2) |
| >5 | 2920 (83.5) | 86 (92.5) |
| Socioeconomic categories* (%) | | |
| Lowest | 1.976 (32.45) | 53 (33.1) |
| Middle | 2077 (34.11) | 53 (33.1) |
| Highest | 2036 (33.44) | 54 (33.8) |
| Distance of HH from nearest Maternal child healthcare centre in km | | |
| <5 | 4882 (80.1) | 125 (78.1) |
| 5–10 | 1550 (18.9) | 29 (18.1) |
| >10 | 60 (1) | 6 (3.8) |

*The scale is a composite of house type, floor, house ownership, separate kitchen, TV, refrigerator, mobile phone, washing machine, total HH income, Number of rooms by principal component analysis.
BPL, Below Poverty Line card; HOH, head of houshold; RSBY, Rashtriya Swasthya Bhima Yojna (Health insurance scheme).

## Place of childbirth

Of the 824 women who gave birth in the previous year, 438 were home births (53.1%, 95% CI (49.1% to 56.6%)) and of the remaining 386, 340 (88%) chose to give birth at a public hospital. Among the women who gave birth in the previous 3 months (n=160), a similar proportion (53.1%, 95% CI (45.0% to 61.0%)) gave birth at home. Only 16.2% (12) of these mothers availed cash incentive through the JSY scheme (table 3). Thirty-six (48%) went to hospital due to initiation of labour pains, 32% due to development of complications and 6.7% reported the reason that they had crossed the expected date of delivery. The individual who was most influential in making the decision of delivering at a hospital was most often (48%) the women herself, followed by the husband (18.7%) or the mother-in-law (17.3%). Irrespective of the place of delivery, only 15% of these households were visited by a health worker within 48 h of delivery and only 30% of women visited a healthcare facility after giving birth. Among those who gave birth in a facility 92% were satisfied with the services provided at the hospital. Women who gave birth at home were more likely to be multiparous, less likely to avail ANC in a public hospital and visit a facility during the postpartum period (table 3).

## Predictors of home births

Among the 824 women who gave birth in the previous year, the following demographic characteristics were significantly associated with home births: living in a rented house, low SES, low literacy of HOH, HOH being an unskilled labourer, migrants and multiparity. Multiparity (OR 1.96, 95% CI (1.44 to 2.69)), literacy status of HOH (OR 0.71, 95% CI (0.53 to 0.97)) and migrant status (1.46, 95% CI (1.08 to 1.97)) remained strong independent predictors of home births in multivariate analysis (table 4).

## Reasons for choosing home birth

The majority of home births were preplanned and 75% of these women had availed some ANC at a facility. Eighty-two per cent of the home births were conducted by a traditional birth attendant (*Dai*). Results from the quantitative and qualitative data showed a high level of concordance in the reasons for choosing home births. Four major themes emerged as barriers to institutional births. Illustrative quotations from the transcripts are presented in box 1.

### Fear and embarrassment

Fear and embarrassment associated with giving birth in hospitals were reported as the most important reasons for giving birth at home during FGDs and IDIs and was also reported as the key reason (35%) among the 85 RDW surveyed. There was fear of being alone in unfamiliar surroundings and fear of surgical intervention. In addition to fear, women felt it was embarrassing and uncomfortable for them to be in the presence of 'strangers' during a very vulnerable time. Lack of privacy coupled with the absence of any family member by their side was in stark contrast to the 'safe and reassuring environment' of their homes during the birthing process.

### Prior experience with hospitals

Prior experience of self, friends, neighbours or a family member played an important role in choosing home or hospital birth. Positive experiences reinforced the message that hospitals were a safe and welcoming place as opposed to negative experiences (such as, perceived improper care and rude behaviour of hospital staff). The healthcare providers interviewed indicated that high patient load at hospitals leads to lack of individual attention and inadequate care.

**Table 3** Information on ANC and births obtained from recently delivered women

| Characteristics* | Home births (n=85) | Institutional births (n=75) | p Value |
|---|---|---|---|
| Mean age | 24.7 (4.31) | 24.9 (4.43) | 0.79 |
| >18 years of age at marriage | 72 (84.7) | 64 (85.3) | 0.912 |
| Mean family size | 5.5 (2.4) | 5.5 (2.3) | 0.937 |
| First child | 15 (17.7) | 26 (34.7) | 0.014 |
| Some ANC care | 63 (74.1) | 72 (96) | <0.0001 |
| ANC at public hospital | 41 (65) | 61 (84.7) | <0.0001 |
| ANC in first trimester | 21 (33.33) | 31 (43.1) | 0.498 |
| Some health problem during pregnancy | 11 (17.5) | 13 (18) | 0.92 |
| Satisfaction with ANC | 46 (74.6) | 65 (90.3) | 0.046 |
| Planned place of birth | 73 (86) | 66 (90.4) | 0.38 |
| Birth conducted by doctor/nurse | 15 (17.7) | 74 (98.7) | <0.0001 |
| Home visitation by community health worker | 14 (16.5) | 11 (14.7) | 0.75 |
| Postpartum visit to hospital | 3 (3.53) | 16 (21.33) | 0.001 |

ANC, antenatal care.
*All continuous variables are expressed as mean and one SD; all proportions are expressed as percentages.

**Table 4** Predictors of home births

| Characteristics (n=824) 438: Home births 386: Institutional delivery | p Value (ignoring clustering) | Crude OR (95% CI), p value (accounted for clustering) | | Adjusted* OR (95% CI), p value (LR test) | |
|---|---|---|---|---|---|
| Buland masjid | <0.001 | 1 | <0.001 | | |
| CPJ | | 0.42 (0.30 to 0.59) | | | |
| Chaderpuri | | 0.69 (0.49 to 96) | | | |
| Birth order second and above | <0.001 | 2.12 (1.57 to 2.87) | <0.001 | 1.96 (1.44 to 2.69) | <0.001 |
| Lower SES | | 1 | | | |
| Middle SES | 0.033 | 0.90 (0.64 to 1.26) | 0.011 | 0.96 (0.66 to 1.406) | 0.68 |
| Highest SES | | 0.70 (0.50 to 0.98) | | 0.91 (0.60 to 1.38) | |
| Joint families | 0.057 | 0.78 (0.57 to 1.05) | 0.102 | | |
| Non-Muslims | 0.004 | 0.77 (0.54 to 1.1) | 0.15 | 0.78 (0.54 to 1.14) | 0.21 |
| Schooling of HOH | 0.007 | 0.74 (0.55 to 0.99) | 0.04 | 0.71 (0.53 to 0.97) | 0.031 |
| Not working | | 1 | 0.063 | 1 | 0.2 |
| Elementary job | 0.024 | 1.63 (1.08 to 2.45) | | 1.54 (1 to 2.38) | |
| Skilled job | | 1.42 (0.92 to 2.17) | | 1.40 (0.89 to 2.21) | |
| Own the house | 0.004 | 0.76 (0.56 to 1.02) | 0.07 | 0.94 (0.65 to 1.34) | 0.72 |
| Ration card possession | 0.037 | 0.80 (0.60 to 1.07) | 0.14 | | |
| Not belonging to Delhi | <0.001 | 1.61 (1.21 to 2.15) | 0.001 | 1.46 (1.08 to 1.97) | 0.013 |

*Adjusted for house ownership, SES, literacy of HOH, Occupation of HOH, belonging to Delhi and birth order.
HOH, head of the household; LR test, likelihood ratio test; SES, socioeconomic status.

## Domestic responsibilities

Being in an unfamiliar neighbourhood, the absence of extended family to help with childcare and traditional lack of involvement of men in childcare made women reluctant to leave children at home and get admitted to a hospital. In the survey, 10% of those who had home births cited lack of help with childcare as a reason.

## Opportunity costs

Although most services at the hospital were provided free of cost, opportunity costs in the form of lost wages for the earning member and cost of food for the family and travel dissuaded some women from delivering in the hospital. Although only 6% mentioned this as a reason in our survey, this emerged as a factor in the qualitative analysis. However, direct costs were not one of the reasons cited for not opting for hospital births.

## DISCUSSION

Our study showed a high prevalence of home births conducted by TBA among the urban poor of north-east district of Delhi. Fear of surgical procedures, unfamiliarity with hospital surroundings, lack of help for childcare and loss of wages were some of the reasons that drove women to choose home births. Other predictors of home births were low literacy, higher parity and migrant status. Concordance between results derived from qualitative and quantitative data lends greater credibility to these findings. Figure 2 presents a conceptual framework based on the study findings which could help us to design strategies for some of the modifiable factors.

The prevalence and reasons for home births in our study were similar to that found in most other urban surveys[13][14][20–26] from India (table 5). In a Mumbai slum study,[25] the prevalence varied from 6% to 16% across 48 slum clusters. Tradition was the most important reason behind home births in this study. Apart from the predictors that we identified, poor housing, lack of water supply and hazardous location were associated with home births in the Mumbai study, indicating that apart from individual and household level factors, the type of neighbourhood also played a role. One of the limitations of our study is that we could not evaluate cluster level predictors of home births as we included only three clusters in this study.

Migrant status was one of the important determinants of home births in our study. In an analysis using NFHS data, Singh et al[27] reported that urban poor migrants were at the highest risk of unsafe birthing practices in contrast to non-poor, non-migrants who were at the least risk. In our study sample almost 60% of the households were migrants from neighbouring states, but most were living in Delhi for more than 5 years. In spite of this, these households were less likely to possess ration, BPL or RSBY cards which are required for availing entitlements to healthcare.

Quality of care did not figure in the discussions as a factor for choosing home births. Quantitative data also confirm that majority of women who visited the hospital for ANC and for birthing were quite satisfied with the services offered. We hypothesise that since the focus on facility-based birthing and offering free services to enable the same is comparatively new to the urban poor community, the community has not reached the stage of assessing the quality of care and using that as a factor in making a decision on where to deliver. Based on our observation of the facilities there is a lot of scope for

---

**Box 1** Illustrative quotations for reasons for home birth

*Fear and Embarrassment*

They prefer home deliveries as in many cases the doctor does not behave well with them. As soon as they enter, they are separated from their families. The doctor does not communicate to the relative if there is any complication. They stay for 1–2 days in the labour room; the relatives are outside they do not know what is happening. It is very scary for them. Also because of all these reasons, they will come only if it is life threatening. Even then they might prefer to go to someone who is local, who is more patient friendly, private practitioner who are non-judgemental who behave properly. They give more one to one care. They might not be qualified but it is more natural and human. Senior gynaecologist, Public health facility

I'd prefer to have the baby at home. If you're in hospital, they don't even attempt to try for a natural birth. What's the use of having an operation if you can have a child the normal way? We get the checkups done there, but we end up having the delivery at home. Pregnant woman

If you tell them to put you in a closed room to get a check-up, they tell you to just lie down and get it done right there. Its humiliating; you can't help but feel embarrassed. And if you don't feel embarrassed, other people around you will. They tell you that if you feel ashamed, go to a private hospital. It's a matter of dignity. There are men walking around as well, if a man catches a glimpse, it can create trouble at home.-Pregnant woman

*Prior experience with hospitals*

I had gone recently with my sister to a hospital……when I went there to deliver my baby, they just kept telling me 'keep pushing, keep pushing…" I got so scared I just left. I've had three children at home; I can manage a fourth the same way. Recently delivered woman

*Other children*

I have small children. If I have to go to the hospital, I have to lock the house and take the children along. Then if she delivers in the hospital, she may be admitted for at least 2 or 3 days depending on the situation. Even if it is a normal delivery it is at least a 2 day stay. How do I manage in such situations? To avoid all this one hopes that if all is well it is better to deliver at home itself. We can all be at home and kids need not have to go anywhere. I can also go for work. Husband of pregnant woman

*Opportunity costs*

Most of them earn daily wages, so they do not want to come to the hospital. They feel one day will go so they will lose their pay also...So the delivery can be at home if baby is okay. Male member does not want to involve himself in all these things, either if there is an elder woman in the house or neighbourhood who conducts the delivery...Even at the hospital they do not want to stay they say that they have to go to work, or how will they earn for tomorrow's food because they are working on daily wages.. Senior gynaecologist, public health facility

improvement in the services being offered suggesting a gap between what level of services women perceive they are entitled to and what they actually receive.

The discomfort of hospitals expressed by those who gave birth at home could be attributed to the overburdening of referral hospitals leading to lack of personalised care. ANC is provided at dispensaries, maternal and child health centres (MCH), secondary level and referral hospitals. However, MCH centres cater to deliveries of multigravida only, and all primigravida are referred to the secondary level or referral hospitals leading to increased patient load at these centres. Initiatives to decentralise care to reduce burden on referral hospitals by upgrading the MCH centres have been rather slow. It might be possible that other supply-side issues could have also contributed to this level of dissatisfaction, but due to delay in obtaining permissions, we were unable to conduct facility assessment to identify the potential causes.

Qualitative data from our study suggest that there was a lack of perceived risk among women and their family members. According to the mothers and the family members who played an important role in decision-making, the practice of giving birth at home was common and that they had witnessed their relatives doing well after doing the same. Priority was given to other domestic responsibilities and tradition over safe births. This may have been particularly relevant among multiparous women.

Low-literacy level of the HOH was another important predictor indicating the need to raise awareness of safe birthing practices. The role of CHWs in improving maternal and neonatal health indicators by increasing awareness is well known.[28] Although the Delhi State Health Mission has deployed Accredited Social Health Activists (ASHA) in the urban areas replicating the rural model, a recent evaluation has shown several implementation gaps.[29]

Afsana and Rashid[30] from Bangladesh report that cost, fear of hospitals due to lack of privacy, unfamiliar surroundings and stigma attached to hospital delivery were key reasons for women to choose home births. While women in our study spoke about fear and embarrassment as deterrents, direct costs associated with delivery and stigma attached to a hospital delivery were not mentioned as factors affecting their choice. Traditionally, Indian women gave birth at home surrounded by close family members. There is evidence to show that the support of a female relative during the birth process is beneficial and leads to better birth outcomes.[31] The state of Tamil Nadu has implemented a successful birth companion scheme through its public health system which addresses the issues of social support, fear and embarrassment.[32] This model provides a prototype that may be replicated, with suitable context-specific modifications, to address this important barrier to institutional deliveries.

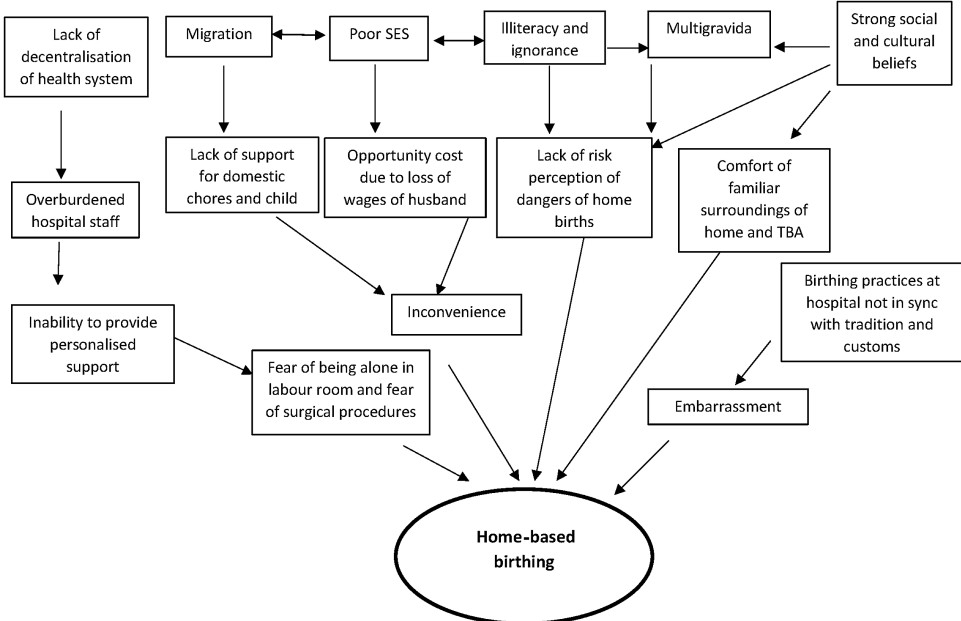

**Figure 2** Conceptual framework of factors leading to home births among the urban poor. SES, socioeconomic status; TBA, traditional birth attendant.

| Table 5 | Prevalence and reasons for home births from urban surveys in India since 2000 | | |
|---|---|---|---|
| Author, year of publication (ref) | Study area and target population/Study design | Sample size and prevalence of home births | Reasons for home births |
| Rahi *et al*, 2006[22] | One Urban slum in Delhi, Births recorded during April–June 2005, cross-sectional survey | n=82 births<br><br>Home births=56.1% | Not reported |
| Agarwal *et al*, 2007[23] | One urban slum in Delhi, women who delivered last 1 year, cross-sectional survey | n=82<br>Home births=31.8% | Lack of awareness for need for check-up (27%)<br>Lack of knowledge about service availability (17%)<br>Long waiting time (22%)<br>None to accompany (15%)<br>Finance (12%)<br>Fear of hospitals (7%)<br>Family objections (2%) |
| DLHS Fact sheet (2007–2008)[14] | Delhi state in 2008 using multistage stratified probability sampling | n=9689 households<br>Home births<br>Rural=42.6%<br>Urban=29.9%<br>Total=30.8% | Not reported |
| Thind *et al*, 2008[20] | NFHS survey data from Maharashtra, cross sectional survey | n=1510 recent births<br>Home births (overall)<br>=37%<br>Only urban=15.3% | Predisposing factors<br>Religion (Hindu), multiple births and caste |
| Agarwal *et al*, 2010[24] | 11 slums of Indore, Madhya Pradesh, cross-sectional survey of mothers of infants (2004–2006) | n=312<br>Home births=56.4% | Not reported |
| Das *et al*, 2010[25] | Mumbai slums from 6 municipal wards, survelllance study (2005–2007) | n=10 754 births<br>Home births=10% | Customary (28%), No time to reach hospital (13%), no body to go along (8%), Fear (7%) |
| Dasgupta *et al*, 2006[21] | Rural and urban clusters in West Bengal from Birbhum district. Cross-sectional survey, women who delivered in the last 1 year | n=320<br>Home births (rural and urban combined)<br>=51.88% | Not reported |
| Khan *et al*, 2009[26] | Periurban area of Aligarh, Uttar Pradesh | n=92 mother of infants<br>Home births=60% | Tradition (42%)<br>Related to economics (31%) |
| Hazarika, 2009[13] | NFHS-3 Delhi data, cross-sectional survey, women who delivered 6 months ago | n=2420 (slum dwellers)<br>Home births=22.62% | Not reported |
| NFHS, National Family Health Survey. | | | |

In our study most births at home were conducted by *dai* who lacked professional training in safe birthing practices. An extensive review by Bergstrom and Goodburn has shown that TBAs had no impact on reduction in maternal mortality.[33] A meta-analysis of training birth attendants showed improved survival but the studies included in the review were from high-mortality burden rural population and not urban population.[34] Koblinsky *et al*[6] analysed national level data from several countries which reduced the MMR drastically since 1950 and showed that maternal deaths could be reduced by providing training to *dais* or professionals developing a partnership with *dais*, however it required apart from political will, effective outreach and referral mechanisms that support traditional system of birthing. Experiences from Malaysia and Sri Lanka show that women are willing to move from home based to facility-based care if transport and services are made free for all, if there is improved awareness and also assured quality of service at the facilities.[6] In India, currently there is no programme at the national scale for promoting or scaling up community-based SBA. The National Rural Health Mission of the Government of India has its main strategy for reduction in maternal mortality focused on facility-based intrapartum care and provision of emergency obstetrics care.

Current initiatives by the government aimed at improving MCH indicators of the urban poor do not directly address some of the key elements identified in our study. Identification and mapping of the most vulnerable populations within the city, sensitisation of health professionals to the needs and fears of women, improving the reach of CHW to the marginalised, empowering women with information regarding their healthcare entitlements, provision of BPL, ration and RSBY cards to the neediest are some of the key issues that need focused and aggressive implementation.

It is important for health departments to strengthen the supply side, be more accessible to those who need them the most and establish faith among the community. India needs to explore innovative ways at all levels of care to make birthing practices safer. There is hope that the urban health situation will improve in the coming years with the NUHM, if we intervene at the individual, community, system and policy levels. The ANCHUL project and similar such endeavours all over the country will need to provide innovative scalable strategies for the betterment of our urban community.

**Author affiliations**
[1]Indian Institute of Public Health-Delhi, Public Health Foundation of India, Gurgaon, Haryana, India
[2]Department of Biochemistry, All India Institute of Medical Sciences, New Delhi, India
[3]Department of Nutrition, Harvard School of Public Health, Boston, Massachusetts, USA

**Acknowledgements** The authors would like to acknowledge all the families who volunteered to provide data for this survey. We thank the efforts of the field staff (field coordinators, field upervisor and field interviewers), data manager and data entry operators.

**Contributors** SGJ and ND conceived and designed the study with additional inputs from AHS and SZ. ND analysed the quantitative data and prepared the first draft of the manuscript. MSG analysed the qualitative data and wrote the first draft of the qualitative findings. GA and AHS supervised data collection and commented on drafts of the manuscript. HN, AHS and SZ commented on drafts of the manuscript. All authors contributed to the critique and modification of the manuscript, read and approved the final version.

**Funding** WHO, Geneva (Alliance for Health Policy and Systems Research (HSS/AHPSR), The Special Programme for Research and Training in Tropical Diseases (IER/TDR), The Special Programme of Research, Development and Research Training in Human Reproduction (HRP), Child and Adolescent Health and Development (FCH/CAH)) (Grant 2011/139172-0)

**Competing interests** None.

**Ethics approval** The study protocol was approved by the Health Ministry Screening Committee of the Government of India, institutional ethics committees of the Public Health Foundation of India, All India Institute of Medical Sciences, WHO Geneva and Harvard School of Public Health.

**Provenance and peer review** Not commissioned; externally peer reviewed.

**Data sharing statement** All unpublished data related to this research project are available with the authors and can be requested by emailing to niveditha@iiphd.org.

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
