## [Reviewer comments · BMJ Open]

Some articles will have been accepted based in part or entirely on reviews undertaken for other BMJ Group journals. These will be reproduced where possible.

ARTICLE DETAILS

TITLE (PROVISIONAL)	Why women choose to give birth at home: A situational analysis from urban slums of Delhi
AUTHORS	Devasenapathy, Niveditha; George, Mathew; Ghosh-Jerath, Suparna; Singh, Archana; Negandhi, Himanshu; Alagh, Gursimran; Shankar, Anuraj; Zodpey, Sanjay

VERSION 1 - REVIEW

REVIEWER	Professor David Ellwood Griffith University, School of Medicine, Queensland, Australia
REVIEW RETURNED	15-Dec-2013

GENERAL COMMENTS	This is a unique study which has been performed very well, using appropriate methodology. The results of this mixed methods approach will greatly inform public health policy in India, especially amongst the urban poor. The coverage of the target population, and the successful completion of a large number of focus groups, is testament to the hard work and energy of the research team and for this they are to be congratulated. I do have one major concern about the approach taken which is the assumption, implicit in the Introduction, that an understanding of why women choose to birth at home will lead to ways of stopping this practice and this will inevitably improve outcomes overall.. That is, it is implicit that birth at home is the cause of the adverse outcome statistics that are presented in the introduction. Is it not more likely that the absence of a trained birth attendant is the main problem? The manuscript would be strengthened overall if there was some further discussion about the reasons behind the maternal and neonatal deaths, and whether or not the birth location was a major factor in all or most of them. A minor point is that the use of the term 'delivery' or 'deliveries' all through the manuscript is not something which would be acceptable in my country. Women give birth (especially at home) rather than 'be delivered' and I would like to see the manuscript altered to reflect this. I also wonder if all of the information about housing, and sanitary facilities is needed as this is not really the focus of the paper. I think some of this material could be cut back without detracting from the overall message of the paper. Thank you for allowing me to read and comment on your work - I think this must have been a lot of very hard work and it is very well written up. I think it will be of great interest to the readers of BMJ Open.
--

REVIEWER	Sabina Faiz Rashid BRAC Univ
REVIEW RETURNED	08-Apr-2014

GENERAL COMMENTS	It does not fully answer the question as to why women give birth at home compared to hospitals. The qualitative data needs to answer some of the questions raised by the title of the paper. The analysis seems to be sporadic and not as well integrated or as rich, whereas the findings from the quantitative seem fine. Do women really not perceive a risk giving birth at home or is do women also view giving birth in hospitals as risky? Why don't they like cesareans? Does it make it difficult for them to work afterwards? What about costs? What is the role of the husband or gatekeepers in hospitals? Other than fear and embarrassment - what about basic quality of care? I believe that if the authors revise the paper and discuss some of these points more clearly it would be a stronger paper and will shed light on women decisions to give birth at home. The references are fine, but better use of other authors who have written anthropological/qualitative research on women delivering at home would be useful (Afsana et al, 2010).
--

VERSION 1 – AUTHOR RESPONSE

Reviewer1:

Reviewer Name Professor David Ellwood

1. This is a unique study which has been performed very well, using appropriate methodology. The results of this mixed methods approach will greatly inform public health policy in India, especially amongst the urban poor. The coverage of the target population, and the successful completion of a large number of focus groups, is testament to the hard work and energy of the research team and for this they are to be congratulated.

We thank the reviewer for his kind comments.

I do have one major concern about the approach taken which is the assumption, implicit in the Introduction, that an understanding of why women choose to birth at home will lead to ways of stopping this practice and this will inevitably improve outcomes overall. That is, it is implicit that birth at home is the cause of the adverse outcome statistics that are presented in the introduction. Is it not more likely that the absence of a trained birth attendant is the main problem? The manuscript would be strengthened overall if there was some further discussion about the reasons behind the maternal and neonatal deaths, and whether or not the birth location was a major factor in all or most of them.

We agree with the reviewer that not all deaths could be solely attributed to home births. Our interpretation of the current evidence suggests some deaths can be prevented by a skilled professional at a home birth, but that addressing the majority of deaths would require facility-based births. The primary causes of maternal deaths globally and in India are post-partum haemorrhage,

obstructed labor and sepsis (Montgomery AL, Ram U, Kumar R, Jha P, for The Million Death Study Collaborators (2014) Maternal Mortality in India: Causes and Healthcare Service Use Based on a Nationally Representative Survey. PLoS ONE 9(1): e83331. doi:10.1371/journal.pone.0083331). These causes are most effectively managed in a facility by a trained professional as they require surgical intervention or rapid administration of IV fluids and antibiotics. In addition, the interventions to be carried out (e.g. blood transfusion, administration of uterotonics as per WHO guidelines, and C-section or assisted delivery) would not be advisable or possible in the home even in the presence of a trained professional. For neonatal deaths the primary causes are prematurity and low birth weight, infection, and asphyxia (Million Death Study C, Bassani DG, Kumar R, et al. Causes of neonatal and child mortality in India: a nationally representative mortality survey. Lancet 2010;376(9755). Again, although home-based neonatal resuscitation can be carried out by a trained professional, the management of these conditions is most effective in a facility setting. Most complications leading maternal or neonatal deaths occur unexpectedly during the intra-partum and immediate post-partum period. As such, while trained professionals may improve outcomes for home births, optimal safe-birth preparedness is currently advocated and fully supported by facility-based birth. In India a dearth of skilled birth attendant at community and facility level is one of the factors contributing to high maternal mortality. A meta-analysis has shown that training birth attendants improved survival but the studies included were only a high mortality burden rural population and not urban population (Wilson A, Gallos ID, Plana N, et al. Effectiveness of strategies incorporating training and support of traditional birth attendants on perinatal and maternal mortality: meta-analysis. BMJ 2011;343). Studies in urban slums (as is the case in the present study) have documented high rate of home births with low attendance by SBA. Most births at home are managed by traditional birth attendants (TBAs). This could be detrimental because of hazardous delivery practices. (Khan Z et al, Indian J Community Med 2009;34(2):102-07) (Agarwal S et al India. J Health Popul Nutr 2010;28(4):383-91). Also , an extensive review by Bergstrom and Goodburn 2001 has shown that TBAs had no impact on any reduction in maternal mortality. Further, in India, states with high rates of home births are those states with high MMR. (Registrar General of India. Sample registration system- Statistical Report 2010. New Delhi: Vital Statistics Division, Ministry of Home Affairs, Government of India; 2010). Hence it is reasonable to implicitly attribute home births to play a big role in contributing to MMR in India. It is shown that if developing and densely populated countries have to move towards drastic reduction in maternal deaths focus should be on health centre based intra-partum care. The lancet series on maternal survival (2006) has shown clearly that Facility based care is effective in improving maternal and neonatal survival. An evaluation study in Indonesia showed that the maternal mortality persisted to be high in two districts compared to the national averages in spite of increasing the availability of trained Birth attendants and skilled professionals as the proportion of home births were very high and care was sought at the facility only in the event of a complication

With this background and the current policy under the National Rural Health Mission of Govt of India which has its main strategy for reduction in maternal mortality focused on institutional deliveries and provision of EmOC, our study was conducted with the intention to inform this policy regarding the factors that may lead to home births in the urban set ups. Identifying barriers at all levels for facility based care in urban set up and working towards these barriers was considered important to inform this policy.

Action: We have added the above points and have referenced to justify our assumption in the background and paragraph 7 of discussion section of the manuscript.

2.A minor point is that the use of the term 'delivery' or 'deliveries' all through the manuscript is not something which would be acceptable in my country. Women give birth (especially at home) rather than 'be delivered' and I would like to see the manuscript altered to reflect this.

We have made the alterations as suggested by the reviewer in the title and the body of the paper.

3. I also wonder if all of the information about housing, and sanitary facilities is needed as this is not really the focus of the paper. I think some of this material could be cut back without detracting from the overall message of the paper.

Yes, we have removed the description on housing, fuel, drinking water, toilet and durable assets from the table 2 and added a line in the description of household characteristics in the results section.

4. Thank you for allowing me to read and comment on your work - I think this must have been a lot of very hard work and it is very well written up. I think it will be of great interest to the readers of BMJ Open.

We thank the reviewer for providing useful inputs in improving the paper.

Reviewer 2:

Reviewer Name Sabina Faiz Rashid

1. It does not fully answer the question as to why women give birth at home compared to hospitals. The qualitative data needs to answer some of the questions raised by the title of the paper.

The analysis seems to be sporadic and not as well integrated or as rich, whereas the findings from the quantitative seem fine.

Thank you for your comments. We have tried to respond to your specific queries below.

1. Do women really not perceive a risk giving birth at home or do women also view giving birth in hospitals as risky?

According to the focus group discussions we had with pregnant women and recently delivered women, they did not consider giving birth at home as risky. According to them, the practice of giving birth at home was common and that they had witnessed their relatives doing well after doing the same. The family (based on the FGD's with mother in laws and fathers) also held this view and they also played a role in deciding the place of birth. It evolved from the discussions that unfamiliarity of the hospital surroundings to be major concern. They did not perceive giving birth at hospital as risky but felt uncomfortable.

Action: We have emphasized these issues in the discussion section. (refer to paragraph 5 of discussion).

2. Why don't they like cesareans? Does it make it difficult for them to work afterwards?

The primary reason stated for dislike for caesareans was fear of the surgery itself. On being probed at their fear, some of the women mentioned that they were worried about being in the 'operation theatre' and they didn't know what would be done to them there. None of them mentioned about difficulty in working afterwards or any other issues as a reason.

Action: We have added a sentence in the section on "Fear and embarrassment" to clarify this in the manuscript.

3.What about costs?

Direct costs were not mentioned anywhere by the women as a reason to deliver at home. We agree that this is in divergence to other studies on access to health care services. However, one of the key informant (A Senior Gynaecologist of a referral hospital) did point out that indirect costs from loss of wages by the husband as an important factor for women to give birth at home.

4.What is the role of the husband or gatekeepers in hospitals?

In general husbands played a minimal role in taking decisions about where the women would deliver and other members in the family especially the mother or in some cases the mother in law or an elder sister who had already had children influenced the decision. Once the woman got admitted into the labour room none of the relatives including the husband had any role or were allowed to be with the woman. This is one of the reasons why women said they were afraid when they are admitted as they felt left alone at a time when they feel the need to be reassured.

As reported by the respondents of the household survey in that area, there was lack of presence of any community health worker who liaised with the hospital or maternity centres who could influence the decision of the woman or her family.

5. Other than fear and embarrassment - what about basic quality of care?

None of the women in our sample spoke about basic quality of care as a determinant in choosing to deliver at home. Fear and embarrassment were the key issues that kept repeating in the various FGDs. Other than the reasons stated above, negative experience with a hospital or its personnel seemed to be a factor in the choice for home birth. Quality of care did not figure in the discussions as a factor that made women or their families choose to birth at home. We hypothesize that since the focus on facility based birthing and offering free services to enable the same is comparatively new to the urban community, the community has not reached the stage of assessing the quality of care and using that as a factor in making a decision on where to deliver. The quantitative data also confirm that majority of women who visited hospital for ANC and for birthing were quite satisfied with the services offered. At the same time, it is clear their fear would suggest perceptions of problems that are not offset by their perceptions of high quality of care or services. Notably, some of those who participated in the discussion were satisfied with the services they received even though exit interviews with this same group of women showed there was a lot of scope for improvement in the services being offered- eg: quality of infrastructure, clean toilets, waiting times for women during ANC check-ups. Again, this suggest a gap between what level of services women perceive they are entitled to, and what they actually receive.

Action: This is now mentioned in the discussion in paragraph 4

6.I believe that if the authors revise the paper and discuss some of these points more clearly it would be a stronger paper and will shed light on women decisions to give birth at home.

We have modified the results and discussion in the paper as shown in track changes highlighting the above points. We have also added a diagram (Figure 2) with all factors as found in this study that was associated with the decision to give birth at home.

7. The references are fine, but better use of other authors who have written anthropological/qualitative research on women delivering at home would be useful (Afsana et al, 2010).

Thank you for directing us to the interesting and relevant article. We have included it in the discussions. (Reference number 30)